# Implementation of QbD Approach to the Analytical Method Development and Validation for the Estimation of Metformin Hydrochloride in Tablet Dosage Forms by HPLC

**DOI:** 10.3390/pharmaceutics14061187

**Published:** 2022-05-31

**Authors:** Mousa Sha’at, Adrian Florin Spac, Iulian Stoleriu, Alexandra Bujor, Monica Stamate Cretan, Mihaela Hartan, Lacramioara Ochiuz

**Affiliations:** 1Department of Pharmaceutical Technology, Faculty of Pharmacy, “Grigore T. Popa” University of Medicine and Pharmacy, 16 Universității Street, 700115 Iași, Romania; mousa-shaat@umfiasi.ro (M.S.); alexandra.m.bujor@umfiasi.ro (A.B.); monica.stamate@umfiasi.ro (M.S.C.); ochiuzd@yahoo.com (L.O.); 2Department of Phisico-Chemistry, Faculty of Pharmacy, “Grigore T. Popa” University of Medicine and Pharmacy, 16 Universității Street, 700115 Iași, Romania; 3Faculty of Mathematics, “Al. I. Cuza” University, 11, Blvd. Carol the 1st, 700506 Iași, Romania; stoleriu@yahoo.com; 4Regional Institute of Oncology, 2-4 General Henri Mathias Berthelot Street, 700483 Iași, Romania; mihaela_hartan@yahoo.com

**Keywords:** metformin hydrochloride, quality by design (QbD), central composite design (CCD), critical analytical attributes (CAA), HPLC, validation

## Abstract

The current studies entail quality by design (QbD)-enabled development of a simple, rapid, precise, accurate, and cost-effective high-performance liquid chromatographic method for estimation of metformin hydrochloride (M-HCl). Design of experiments (DoE) was applied for multivariate optimization of the experimental conditions of the HPLC method. Risk assessment was performed to identify the critical method parameters (CMPs) using Ishikawa diagram. The factor screening studies were performed using a two-factor three-levels design. Two independent factors, buffer pH and mobile phase composition, were used to design mathematical models. Central composite design (CCD) was used to study the response surface methodology and to study in depth the effects of these independent factors, thus evaluating the critical analytical attributes (CAAs), namely, retention time, peak area, and symmetry factor as the parameters of method robustness. Desirability function was used to simultaneously optimize the CAAs. The optimized and predicted data from contour diagram consisted of 0.02 M acetate buffer pH = 3/methanol in a ratio of 70/30 (*v*/*v*) as the mobile phase with a flow rate 1 mL/min. The separation was made on a Thermoscientific ODS Hypersyl^TM^ chromatographic column (250 × 4.6 mm, 5 μm) with oven temperature 35 °C and UV detection at 235 nm. The optimized assay conditions were validated according to ICH guidelines. Hence, the results clearly showed that QbD approach could be successfully applied to optimize HPLC method for estimation of M-HCl. The method was applied both for the evaluation of M-HCl content in tablets, and for in vitro dissolution studies of M-HCl from conventional and prolonged-release tablets.

## 1. Introduction

Metformin (1,1-dimethylbiguanide or N,N-dimethylimidodicarbonimidic diamide, C_4_H_12_ClN_5_) has a molecular weight of 129.16 and a melting point at 223–226 °C. Metformin hydrochloride (M-HCl, Figure 1) is the hydrochloride salt of the biguanide metformin, used to treat high blood sugar levels that are caused by a type of diabetes mellitus.

In the literature, research studies have been identified in which metformin has been linked to a possible reduction in the risk of cancer [1,2], but also to the fact that it also reduces mortality caused by oncological diseases, patients being protected from various types of cancer, such as colon, gastric, breast, endometrial, and glioma [3,4,5,6,7].

Pharmacologic therapy for Type 2 diabetes involves use of metformin as first-line therapy. Metformin hydrochloride acts through three mechanisms; primarily, it reduces hepatic glucose synthesis (inhibits gluconeogenesis and glycogenolysis), at the muscle level, is responsible for increasing insulin sensitivity by intake and using it, but it also delays intestinal glucose absorption [8].

Metformin used alone has been shown to reduce glycosylated hemoglobin [9] by approximately 1.5% and is currently available as 500 mg, 850 mg, and 1000 mg immediate release (IR); extended-release tablets (XR) of 500 mg, 750 mg, and 1000 mg; 500 mg/5 mL liquid formulation, or 500 mg powder sachets [10]. New studies on new pharmaceutical formulations of metformin show major benefits in lipid metabolism and improved glycemic control in patients who have used the prolonged-release form. Reducing the frequency of one tablet per day in the case of metformin XR results in better gastrointestinal tolerability through gradual and controlled release of the active substance, increased compliance with therapy by reducing the number of doses administered per day, but also reducing the number and intensity side effects reported.

In the literature there are presented different methods for metformin hydrochloride analysis in various pharmaceutical formulations, synthetic mixtures, water samples, human plasma, biota, and others, by microgravimetry, electrochemical methods (such as conductometric, voltammetric, and potentiometric), spectrofluorimetry, UV–Vis spectrophotometry, capillary electrophoresis, gas chromatography, and HPLC. The HPLC methods are used for the qualitative identification and quantitative determination of M-HCl alone or in combination with other antidiabetic class, such as sulfonylureas derivatives (i.e., gliclazide, glipizide, glimepiride glibenclamide and tolbutamide); thiazolidinediones (glitazones) (i.e., pioglitazone); alpha-glucosidase inhibitors (i.e., miglitol, acarbose, and voglibose); methylglinides (i.e., nateglinide and repaglinide); dipeptidyl peptidase-4 (DPP-4) inhibitors (i.e., sitagliptin, saxagliptin, vildagliptin, alogliptin, and linagliptin); sodium-glucose co-transporter 2 (SGLT-2) inhibitors (i.e., dapagliflozin, canagliflozin, empagliflozin, ertugliflozin pidolate, and remogliflozin) or with other therapeutic classes of molecules (hypocholesterolemic) [11,12,13,14,15,16,17,18,19,20,21,22,23,24,25,26,27,28,29,30,31,32,33,34].

The quality by design (QbD) approach suggests looking into the quality of the analytical process during the development stage itself. It says that quality should be built into the process design rather than testing final results of analytical process. QbD is defined as a systematic approach to development that begins with predefined objectives and emphasizes product and process understanding based on sound science and quality risk management [35,36]. In alignment with the approach proposed in the draft FDA guidance for process validation [37], a three-stage approach can be applied to method validation: Stage 1. Method Design: define method requirements and conditions and identify critical controls; Stage 2. Method Qualification: confirm that the method is capable of meeting its design intent; Stage 3. Continued Method Verification: gain ongoing assurance to ensure that the method remains in a state of control during routine use.

The QbD approach is more appropriate for application during method development than during method validation, because method validation is a process which demonstrates that the analytical method is appropriate for its intended use. In the development of analytical methods it is frequent practice to implement the principles of QbD. These principles facilitate the scientific and risk-based understanding of major sources of variation. In this way, a high robustness and an improved performance of the analytical methods are obtained.

Regulatory agencies do not define any specific process of analytical QbD; however, a parallel approach can be drawn based on product QbD. Equivalent to process QbD, the outcome of analytical QbD (AQbD) is well understood and fit for intended purpose with robustness throughout the lifecycle. Quality target product profile (QTTP), critical quality attributes (CQA), and design of experiments (DOE) can be interpreted as analytical target profile (ATP), critical method attributes (CMA), such as retention time, peak area, symmetry factor, tailing factor, the resolution between adjacent peaks, plate count, etc., and method operable design region (MODR), respectively [38,39].

Fundamentally, the QbD analytical approach requires the identification of the ATP before considering analytical technology. The next step is to establish the CMAs. An MODR is required for analytical methods during the development phase for a better improvement as well as for a quantitative understanding of the factors that influence the performance of the method. To find high-risk variables that have a critical impact on analytical performance, the aim is to identify critical method parameters (CMPs) such as material attributes, instrument-related aspects, instrument operating parameters, and method parameters, based on risk assessment and factor screening studies, followed by their optimization using appropriate experimental projects to increase method performance. The optimized chromatographic conditions are thus evaluated in terms of the CMAs based on the knowledge obtained in the development stage and the screening studies [35,38,40,41,42].

An important aspect for the development of the HPLC method using the QbD approach using preliminary risk assessment experiments is the choice of CMPs and responses. In addition to the preliminary experiments, the Ishikawa (fish bone) diagram was used to identify and evaluate the CMPs that pose a risk to the performance of the method.

Each analytical technique has different CMPs. In the case of HPLC methods, the CMPs with high risk assessment are those regarding the material, instrumental parameters, mobile phase (buffer type, concentration, and pH, organic modifier, elution method), column characteristics and preparation, and the analyst. In general, for risk identification and assessment, Ishikawa fishbone diagram can be used (Figure 2) [42,43,44,45,46].

In short, the necessary steps in developing an analytical method with the implementation of QbD principles are as follows: Project initiation → Literature search and initial risk assessment → Identification of ATP, CMAs, risk assessment → Method optimization and method development with DOE → MODR, Control Strategy and Risk Assessment → AQbD method validation → Continuous method monitoring.

With all these in mind, the aim of this work was to go through the necessary steps, namely, the development of an HPLC method for the quantitative determination of M-HCl and its optimization in accordance with QbD principles. The next steps were to validate the method according to ICH Q2 (R1) Guidelines [47] and to apply the optimized and validated HPLC method for the quantitative determination of M-HCl from tablets and for dissolution studies.

## 2. Materials and Methods

### 2.1. Chemicals and Reagents

Metformin hydrochloride (97%) was purchased from Sigma Aldrich Chemie Gmbh (Steinheim, Germany), sodium acetate (CH_3_COONa, ≥99.0%) was obtained from Silal Trading SRL (Bucharest, Romania), sodium phosphate dibasic (Na_2_HPO_4_, ≥99.0%) was obtained from Sigma-Aldrich Chemie GmbH (Steinheim, Germany), potassium dihydrogen phosphate (KH_2_PO_4_, ≥99.5%) was obtained from Utchim SRL (Râmnicu Vâlcea, Romania), potassium chloride (KCl, ≥99.0%) was obtained from Chemical Company S.A. (Iasi, Romania), orthophosphoric acid (H_3_PO_4_, ≥85%) was obtained from Chemical Company S.A. (Iasi, Romania), glacial acetic acid (CH_3_COOH, 99.84%) was obtained from Chimreactiv SRL (Bucharest, Romania), hydrochloric acid (HCl, ≥37.0%) was obtained from Chemical Company S.A. (Iasi, Romania), sodium hydroxide (NaOH, 98.5%) was purchased from Chemical Company S.A. (Iasi, Romania), gradient grade methanol and acetonitrile (Hiposolv Chromanorm) were obtained from VWR International S.A.S. (Fontenay-sous-Bois, France). Ultrapure water (resistivity 18.2 MΩ·cm) was obtained from a local pharmaceutical company. Industrial M-HCl tablets with a content between 500 mg and 1000 mg were purchased from the local market (pharmacy).

### 2.2. Equipment

Agilent Technologies 1200 liquid chromatograph was equipped with quaternary pump (type G1311A), diode array detector (DAD type G1315B), degasser (type G1322A), oven column compartment (type G1316A), and Agilent ChemStation 32 software (Rev. B.03.02) (all from Agilent Technologies, Santa Clara, CA, USA). The used chromatographic column was a Thermoscientific ODS Hypersyl TM (250 × 4.6 mm, 5 μm), Lithuania. Other instruments used were pH-meter, inoLAB pH 7110 (Xylem Analytics Germany GmbH, Weilheim, Germany), water bath Biobase (model SY-1L4H, Biobase Biodustry, Shandong, Co., Ltd., Jinan, China), ultrasonic bath Biobase (model UC-40A, Biobase Biodustry, Shandong, Co., Ltd., Jinan, China), analytical balance PIONEER^®^ Analytical OHAUS PX124M (Ohaus Corporation, Parsippany, NJ, USA), dissolution apparatus SR 8 Plus Dissolution Test Station (model 73-100-104, Hanson Research, Chatsworth, CA, USA), microliter™ Syringes, 20 μL Hamilton Bonaduz AG (CH-7402 Bonaduz, Switzerland), Transferpette^®^ Dig. 100–1000 μL (article 704180, Brand GMBH + CO KG, Wertheim, Germany) and Rotilabo^®^-Mikroliterpipette 0.5–5.0 mL (article TA 26.1, Carl Roth GmbH, Karlsruhe, Germany).

### 2.3. Preparation of Standard Solutions

Standard solutions were obtained by dissolving 103.1 mg of M-HCl (reference substance, 97%) in 25 mL of ultrapure water (resistivity 18.2 MΩ·cm) and diluting with ultrapure water after complete dissolution to 50 mL to give a solution with a concentration of 2 mg/mL (2000 μg/mL). Any further dilutions for standard solutions were also performed with ultrapure water.

### 2.4. Selection and Preparation of Mobile Phase

Mobile phases containing methanol, acetonitrile, water, and buffers at different pH were tested in different proportions. Peaks with satisfactory CMAs (retention time, peak area, symmetry factor) were obtained at a flow rate of 1 mL/min with a mobile phase made up of 20 mM acetate buffer (pH 3)/methanol 70/30 (*v*/*v*). To prepare a volume of 1 L of mobile phase, a volume of 300 mL methanol is mixed with 700 mL 0.02 M acetate buffer (pH = 3). After the mixture reached room temperature, the mobile phase was filtered through 0.45 μm membrane filters by application of vacuum and sonicated for 15 min before introducing into the system.

### 2.5. Method Development and Optimization Using QbD Approach

#### 2.5.1. Identification of ATP, CMAs, and Risk Assessment

Identification of ATP was made after literature search and initial risk assessment in which were followed studies that present different methods for determining M-HCl by HPLC, respectively, the implementation of QbD principles in the development of HPLC methods. The selected CMPs were related to a different composition of the mobile phase (i.e., buffer type, buffer pH, organic modifier type, and buffer to organic modifier ratio). For risk identification and assessment, Ishikawa fishbone diagram was used (Figure 2).

#### 2.5.2. Method Development and Optimization

In order to obtain an adequate separation of M-HCl with acceptable values of system suitability parameters (retention time, peak area, and symmetry factor), the chromatograms obtained by HPLC were recorded for various compositions and mobile phase ratios using various buffer solutions (KH_2_PO_4_ and CH_3_COONa acetate buffer with different pH values) and organic modifier (methanol and acetonitrile).

To study the influence of the aqueous phase and of the organic modifier on the separation performance, we first use as a mobile phase a mixture of methanol with water, phosphate buffer, and acetate, respectively (with the same concentration of 0.02 M and pH = 3), and secondly, methanol was replaced with acetonitrile.

To study the influence of the buffer pH and the ratio between the buffer solution and the organic solvent in the composition of the mobile phase, we choose a central composite design (CCD) in which the star points are at the center of each face of the factorial space [46,48,49,50]. This variety requires three levels of each factor (–1, 0, +1). As independent variables (factors selected based on the preliminary analysis), the composition of the mobile phase and the pH of the buffer were chosen. In Table 1 is presented the design matrix with the selected factors at low (–1), medium (0), and high (+1) levels for a number of 11 experimental runs with triplicate tests for the central point (0, 0). The dependent variables were retention time, peak area, and symmetry factor for the proposed independent variables. For the dependent variables, the used constraints were minimum for retention time and maximum for peak area and symmetry factor.

Experimental design (a two-factor, mobile phase composition and pH of buffer solution at three different levels), desirability function, and data analysis calculations were performed by using MATLAB and Statistics Toolbox Release 2020a (The MathWorks, Inc., Natick, MA, USA) software, the best-suited response for second-order polynomial exploring quadratic response surfaces [40,46]:Y = β_0_ + β_1_A + β_2_B + β_3_AB + β_4_A^2^ + β_5_B^2^(1)
where A and B are independent variables coded for levels, Y is the measured response associated with each combination of factor level, β_0_ is the intercept, and β_1_ to β_5_ are regression coefficients derived from experimental runs of the observed experimental values of Y. Interaction and quadratic terms, respectively, are represented by the terms AB, A^2^, and B^2^. The surface response of dependent variables and the desirability plot were also plotted using MATLAB software.

#### 2.5.3. Risk Assessment and Control Strategy

The optimized method is checked by the CMAs to see if the method is efficient and operational throughout its life. Thus, for robustness and ruggedness studies, the parameters and performance of the method were evaluated in several deliberately obtained circumstances (different reagents, analysts, and days). The robustness of the method was determined by making some minor changes in the CMPs (source of methanol, mobile phase flow, and pH of the buffer solution) and the ruggedness was determined by changing the analyst and the days when analyses were performed. The relative standard deviation (RSD%) acceptance limits for retention time, peak area, and symmetry factor must be less than 2%.

Following the development of the method, a control strategy was implemented for the development of which the ATP was established, i.e., a planned set of controls of some parameters to ensure that both the quality of the results obtained and the method performance fall within the established ATP.

### 2.6. Method Validation

The HPLC method for M-HCl was validated in terms of linearity, limit of detection (LOD), limit of quantification (LOQ), precision, and accuracy according to ICH Q2 (R1) Guidelines [47].

The linearity was determined by preparing a calibration curve for 19 standard solutions with concentrations ranging from 10 to 2000 μg/mL. Each solution was analyzed in triplicate; from the obtained chromatograms, the peak areas were determined by integration and used to generate the calibration curve using the corresponding concentration of M-HCl. The equation of regression line was determined using the least squares method and mathematical estimates of the degree of linearity (correlation coefficient—r, coefficient of determination—r^2^, the slope and the intercept with ordinate of the regression line), using regression function in Excel.

The detection limit (LOD) and quantification limit (LOQ) were calculated using the equations *Signal*/*Noise* > 3 and *Signal*/*Noise* > 10 for LOD and LOQ respectively, where signal is the peak area for the signal and noise is the peak area for noise. These limits are calculated using the equation of the regression curve obtained in linearity study:(2)$$LOD=\frac{3\times Area_{Noise}-Intercept}{Slope}\;\mathrm{and}\;LOQ=\frac{10\times Area_{Noise}-Intercept}{Slope}$$

The precision of the developed HPLC method was evaluated in terms of repeatability (intra-day precision) and intermediate precision (inter-day precision). For determination of repeatability, we evaluated in triplicate, on the same day, three solutions of M-HCl at three concentrations (700, 1000, and 1300 μg/mL). Similarly, for the determination of intermediate precision, the above-prepared solutions were analyzed on three consecutive days. The repeatability and intermediate precision of the method were presented as RSD%.

The accuracy of the method has been determined by application of the analytical procedure to recovery studies using the standard addition method. In this, formerly evaluated sample solutions consisting of a known amount of M-HCl were spiked with three different concentration levels. Briefly, to 0.6 mL of 1000 μg/mL M-HCl solution, 0.4, 0.7, and 1.0 mL of 2000 μg/mL M-HCl and 1.0 mL, 0.7 mL, and 0.4 mL of water were added to obtain a final volume of 2 mL. In this way, solutions with concentrations of 700, 1000, and 1300 μg/mL M-HCl were obtained. The accuracy was expressed in terms of percent recovery for final concentrations. The accuracy of the method was also studied at lower concentrations, in this case being calculated in terms of percent relative error (RE%) using the following equation:(3)$$\mathrm{RE}\%=\frac{\left|C_{M}-C_{R}\right|}{C_{R}}\times 100$$
where *C_M_* and *C_R_* are the measured and real concentration, respectively.

### 2.7. Stability Studies for M-HCl Solutions

For the stability study over time of M-HCl solutions, we prepared solutions at concentrations of 1000 and 2000 µg/mL. These solutions were analyzed initially and then after 24 and 48 h, the solutions were stored in the refrigerator, at room temperature, and in a water bath at 37 °C. In all cases, recovery of M-HCl was calculated.

### 2.8. Determination of M-HCl Content in Tablets

Initially, the average weight of one tablet (*M_m_*) was determined according to the provisions of the European Pharmacopoeia, 10th edition [51], for a number of 20 tablets, then the 20 tablets were crushed into powder form. Samples of formulated M-HCl (tablets) were prepared by dissolving a quantity of the powdered tablets equivalent to 500 mg of M-HCl in 500 mL ultrapure water to obtain a concentration of M-HCl of 1000 µg/mL. The obtained solution was filtered through 13 mm polytetrafluoroethylene (PTFE) filters, 0.45 µm, analyzed in the method conditions, and the percent content (C%) of M-HCl was calculated using the following equation:(4)$$C\%=\frac{M_{m}}{A\cdot a}\cdot \frac{P_{A}-Int}{S}\cdot 50$$
where *M_m_* is average weight of a tablet calculated for 20 tablets (in g), *A* is the declared content (in mg), *a* is the quantity of tablet powder (in g), *P_A_* is the peak area, and *Int* and *S* are the intercept and the slope of the regression line, respectively.

### 2.9. Dissolution Studies

In vitro dissolution tests were performed according to the specifications of the “2.9.3. Dissolution Test for Solid Pharmaceutical Forms” and “5.17. Recommendations on methods for dosage forms testing” of the European Pharmacopoeia, 10th edition [51].

The dissolution tests were performed at 37 ± 0.5 °C, using apparatus 2 (paddle apparatus). Two dissolution media were prepared for dissolution studies: for simulated gastric fluid with pH = 1.2 (3.7 g KCl, 7.5 mL concentrated HCl, and distilled water up to 1000 g) and simulated intestinal fluid pH = 6.8 (6.8 g KH_2_PO_4_, 22.4 mL 1M NaOH solution, and distilled water up to 1000 g). The pH value of the dissolution media was checked with a pH-meter and, if necessary, the solution was adjusted with concentrated HCl or 1 M NaOH solution, as appropriate. The test sample (tablet) was placed on the bottom of the cylindrical vessel, after which the air bubbles were removed from the surface of the test sample; the apparatus was started, and the rotational speed was adjusted. The dissolution test for conventional release tablets (tablets and film-coated tablets, samples named CP-2, CP-3, and CP-4) was performed for 2 h under the following working conditions: dissolution medium 500 mL solution for simulated gastric fluid with pH = 1.2; temperature 37 ± 0.5 °C; 60 rpm; duration 2 h. In the case of prolonged-release tablets, named CP-1 and CP-5, the dissolution test was performed for a period of 24 h, using specific working conditions, as follows: first 2 h, simulated gastric fluid pH = 1.2; temperature 37 ± 0.5 °C; 60 rpm. The dissolution medium was replaced with simulated intestinal fluid pH = 6.8; temperature 37 ± 0.5 °C; 60 rpm. A total of 2 mL of medium was collected; after each sampling, volume was kept constant in the cylindrical vessel by replacing with the same volume of fresh dissolution medium at 37 °C. In the case of conventional release tablets, we took samples at the following intervals: 5, 10, 15, 30, 45, 60, 90, and 120 min; for prolonged-release tablets, we took samples on time: 1, 2, 3, 4, 5, 6, 7, 8, 9, 10, 11, 12, and 24 h. Samples were collected at the declared time from the distance between the surface of the dissolution medium and the paddle, but also at least 10 mm from the wall of the vessel, then filtered using nylon filters (0.45 μm; diameter 25 mm). The samples were analyzed by the described method. If the area of the peak corresponding to M-HCl was greater than that obtained for standard M-HCl (1300 μg/mL), the solution was diluted twice and reanalyzed, from where a dilution factor (DF) appeared in the equation used for calculations. The dissolution test was performed on six samples (tablets).

The following equations were used to determine the amount of M-HCl released:(5)$$C_{I}\%=DF\cdot \frac{P_{A\left(tx\right)}-Int}{S}\cdot \frac{500}{1000}\cdot \frac{100}{A}$$
(6)$$C_{II}\%=DF\cdot \frac{P_{A\left(tx-1\right)}-Int}{S}\cdot \frac{500}{1000}\cdot \frac{2}{500}\cdot \frac{100}{A}$$
(7)$$C\%=C_{I}\%+C_{II}\%$$
where*C%* = percentage release in the dissolution medium;*C_I_%* = percentage concentration calculated for the first sampling;*C_II_%* = percentage concentration calculated in the 2 mL taken previously;*DF* = dilution factor (1 or 2);*P_A_* = peak area (mAU·min);*Int* and *S* = intercept and slope of the regression line respectively;*A* = declared content (mg);*tx* = current sampling time;*tx* − 1 = previous sampling time.

At simulated gastric fluid with pH = 1.2, Equation (5) was used for the first sample taken, at the time of 5 min (conventional release tablets) and 1 h (prolonged-release tablets); for the following samples, Equation (7) was used. In the case of prolonged-release tablets, the dissolution medium was replaced with simulated intestinal fluid pH = 6.8, so that Equation (5) was used for the three hour sample, then Equation (7) for subsequent sampling. The values obtained were added with the final concentration obtained at medium pH = 1.2 (simulated gastric fluid).

## 3. Results and Discussions

### 3.1. Method Development and Optimization Using QbD Approach

#### 3.1.1. Identification of ATP, CMAs, and Risk Assessment

For the assay and dissolution study of M-HCl from tablets using the HPLC technique with UV detection, the desired ATP was to obtain a retention time of less than 10 min, and the area and symmetry of the peak to be as large as possible. The CMAs requirements are to use a simple mobile phase (buffer/organic modifier) with isocratic elution for an aqueous sample with a concentration around 1000 μg/mL using a C18 chromatographic column with UV detection with a minimum retention time, with maximum peak area and symmetry of the corresponding peak. A last, but not least, important attribute is that the HPLC separation method can also be used for mass spectrometry (MS) detection.

For HPLC analysis, most research studies report the use of a non-polar stationary phase (C18) in different chromatographic columns (lengths between 50 and 250 mm, inner diameter of 3.9 or 4.6 mm, and particle size of 4 or 5 μm) and a mobile phase that consists of a mixture of two or three solvents (e.g., acetonitrile, methanol, water, phosphate buffer, ammonium buffer, tetrahydrofuran), with aqueous phase having a pH = 2.8–7.0. The ratio of mobile phases differs greatly, for example, the organic modifier being used in a proportion of 5–65%. The flow rate of the mobile phase was generally 1 mL/min, with extreme values of 0.5 mL/min and 1.7 mL/min. The detection was also different (by mass spectrometry or by spectrophotometry in UV). For most research studies, the detection was performed in UV, the wavelength used being between 221–270 nm, most of which were between 220 and 235 nm) [11,12,13,14,15,16,17,18,19,20,21,22,23,24,25,26,27,28,29,30,31,32,33,34].

Regarding the CMPs, from Ishikawa diagram and preliminary experiments that were conducted, CMPs selected for the further study with high risk assessment that can cause variability are the instrument precision and the mobile phase (buffer type, buffer pH, type of organic modifier, and the ratio between the buffer and the organic modifier). Different compositions of the mobile phase (i.e., buffer type, buffer pH, type of organic modifier, and the ratio between the buffer and the organic modifier) were used to study the influence of the mobile phase on the results. To study the precision of the instrument, after the final separation conditions were established, the same solution was analyzed six times and the results were calculated as RSD% of the peak area. The obtained value was less than 2% (see Section 3.2.3).

#### 3.1.2. Method Development and Optimization

Once the CMPs were identified, the next step was to optimize them in terms of the CMAs. As shown before, in order to better understand the performance of the method and to identify the independent CMPs and their effect on the dependent variables, various preliminary experiments were performed by trial and error.

##### Effect of Chromatographic Factors on Responses

Initially, a mobile phase formed by water/methanol (80/20 *v*/*v*) was tried; the peak was observed to be asymmetric. The further mobile phase tried was 0.02 M KH_2_PO_4_ (pH = 3) or 0.02 M CH_3_COONa buffers (pH = 3)/methanol (80/20, *v*/*v*). It was observed that the improvement of peak shape and symmetry was achieved by adjusting the buffer pH. In both cases, the retention times are close to each other, but when the acetate buffer is used, the peak area increases by about 8% and the peak height decreases by about 7.5%, which leads to an increase in detection sensitivity.

Taking into account the fact that one of the purposes of the method is to be able to be also used for mass spectrometry (MS) detection, we chose to use acetate instead of phosphate buffer.

Next, the nature of the organic modifier in the composition of the mobile phase was studied. For this, we replaced the methanol in the mobile phase with acetonitrile, with the percentage and pH of the buffer solution being kept constant (70% and pH = 3, respectively). In this case, there is no significant change in retention time and peak symmetry, but it is observed that a hypochromic effect occurs, decreasing the height and area of the peak, which leads to a decrease in detection sensitivity. As a conclusion, we kept methanol in the mobile phase composition.

##### Evaluation of Experimental Results and Selection of Final Method Conditions

For the study of the influence of the pH value of the buffer solution and of the ratio between buffer solution and the organic solvent in the composition of the mobile phase, we chose a face CCD. Using the CCD approach, these method conditions were assessed. At the first step, the conditions for retention time, peak area, and symmetry factor were evaluated. For M-HCl, this led to distinct chromatographic conditions. The acceptable value falls within those regions where deliberate variations in the parameters of the method do not affect the quality of HPLC separation. On the basis of the factor screening studies, selection of the CMPs actually affecting the method performance was optimized using a two-factor CCD at three equidistant levels, i.e., low (−1), intermediate (0), and high (+1) levels. Table 2 summarizes the design matrix as per the CCD. The same standard concentration was used for all experimental runs, which were analyzed for method CMAs, i.e., retention time, peak area, height, tailing, and symmetry.

As an example, Figure 3 shows the chromatograms obtained for a mobile phase composition consisting of (a) 0.02 M acetate buffer (pH = 3)/methanol in a ratio of 70/30, 80/20, and 90/10, *v*/*v*, and (b) 0.02 M acetate buffer (pH = 3, 4, and 5)/methanol in a ratio of 70:30, *v*/*v*.

##### Design Space

The response surface study type, a CCD with 11 runs, was used. The proposed CCD experimental design was applied and the evaluation of mobile phase composition and pH of buffer was performed against the three responses, retention time, peak area, and symmetry factor; the results are summarized in Table 2 and Figure 4, respectively.

After calculation of a second-order polynomial exploring quadratic response surfaces, in the models below, we have retained only those coefficients β that are significant at 95% confidence level. The results are presented in Table 3.

From the equation for the retention time (*Rt*), that is, *Rt* = 10.209 *+* 0.38833 × *A* − 0.19623 *× B +* 0.0015567 *× B*^2^, we see that the coefficient of A is positive (+0.38833). We can interpret this coefficient as follows: if we keep B fixed, then an increase/decrease in A by one unit will determine an increase/decrease in retention time by 0.38833 units. On the other hand, the coefficient of B is negative (−0.19623), but this coefficient can no longer be interpreted as we did before, because the term *B*^2^ also appears in the formula. We can only say that, for a fixed value of A and the range of B between 70 and 90, retention time is an increasing function of B. This means that a decrease in B will determine a decrease in retention time.

From the equation for the peak area (*Pa*), that is, *Pa* = 4979.6 − 736.18 × *A* − 15.384 × *B* + 70.254 × *A*^2^, we see that the coefficient of B is negative (−15.384). We can interpret this coefficient as follows: if we keep A fixed, then an increase/decrease in B by one unit will determine a decrease/increase in peak area by 15.384 units. We also see that the coefficient of A is negative (−736.18), but this coefficient can no longer be interpreted as we did for B, because the term A^2^ also appears in the equation. We can only say that, for a fixed value of B and the range of A between 3 and 5, peak area is a decreasing function of A. This means that a decrease in A will determine an increase in peak area. In other words, if we intersect the peak area response surface by the plane of equation B = c (here, 70 ≤ c ≤ 90), then the curve of intersection represents a decreasing function of A. Therefore, a decrease in A will determine an increase in peak area.

Similarly, from the equation for symmetry factor (*Sf*), that is, *Sf* = 3.876 − 0.0667 × *A* − 0.0692 × *B +* 0.00038667 × *B*^2^, we see that the coefficient of A is negative (−0.0667). We can interpret this coefficient as follows: if we suppose that B is fixed, then an increase/decrease in A by one unit will determine a decrease/increase in symmetry factor by 0.0667 units. The coefficient of B is negative (−0.0692), but this coefficient can no longer be interpreted as we did for A, because the term B^2^ also appears in the equation. We can only say that, for a fixed value of A and the range of B between 70 and 90, symmetry factor is a decreasing function of B, and, thus, a decrease in B will determine an increase in symmetry factor.

In conclusion, from Figure 4a–c and the equations for the retention time, peak area, and symmetry factor, we can conclude that if both the pH value (code factor A) and the percentage of sodium acetate buffer (code factor B) in the mobile phase composition decrease, then the value of the retention time decreases simultaneously with the increase in peak area and the increase in symmetry factor for the corresponding M-HCl peak.

##### Optimized Chromatographic Conditions

The search for the optimal solution was performed by numerical optimization by “trading off” various CAAs to achieve the desired objectives, i.e., maximization of peak area and symmetry factor and minimization of retention time, to obtain the desirability function close to 1. The optimized solution showed that the mobile phase composition containing a mixture of 0.02 M acetate buffer (pH = 3)/methanol in a ratio of 70/30 (*v*/*v*) yielded desirability close to 1.0, along with all the CAAs in the desired ranges. The optimized values for pH and buffer content in the mobile phase and predicted responses are shown in Table 4. The 3D desirability plot is presented in Figure 5.

Thus, the optimal mobile phase consists of a mixture of 0.02 M acetate buffer (pH = 3)/methanol in a ratio of 70/30, *v*/*v*, conditions in which the method is faster and has a higher sensitivity. The final chromatographic conditions for M-HCl are shown in Table 5. 

As a comment, if a higher percentage of organic modifier is used (for example, for a mobile phase consisting of 0.02 M CH_3_COONa (pH = 3)/CH_3_OH in a ratio of 60/40, the method is faster (retention time decreases) and slightly more sensitive (peak area increases). However, we opted for a 70/30 ratio for three reasons: (i) at a percentage lower than 70% of the buffer solution in the composition of the mobile phase, a negative peak appears in front of the corresponding peak of M-HCl, which makes the integration difficult, and the precision of the peak integration decreases; (ii) reducing costs (by reducing the amount of methanol used; and (iii) reducing pollution (by disposing of as little organic solvent as waste).

#### 3.1.3. Risk Assessment and Control Strategy

For robustness and ruggedness studies, a solution of M-HCl with a concentration of 1000 μg/mL was used. The robustness study was performed by deliberately changing the value of CMPs (methanol from another source, mobile phase flow between 0.9 and 1.1 mL/min, and pH of the buffer solution between 2.8 and 3.2). For the ruggedness study, the determinations were made on different days by another analyst. In both cases, the values obtained for the retention time, peak area, and symmetry factor, respectively, had RSD values less than 2%.

As a control strategy, we planned a set of controls of some parameters to ensure that both the quality of the results obtained and the method performance fall within the established ATP (preparation and storage conditions of samples, measurements performed, and doubling of control operations).

### 3.2. Method Validation

#### 3.2.1. Linearity

Linearity of the proposed method was evaluated according to the ICH guidelines [47]. A number of three sets of working solutions were prepared for the study of linearity over the concentration range 10–2000 μg/mL. Each of these samples was analyzed under the mentioned conditions and, from the chromatograms obtained, the area of the peaks corresponding to M-HCl was determined. M-HCl showed linearity in the concentration range of 10–2000 μg/mL (r^2^ = 0.9999). The regression equation obtained was *P_A_* = 2.4266 × *C* − 56.361, where *P_A_* is peak area and *C* is concentration of M-HCl in μg/mL. 

#### 3.2.2. Limit of Detection (LOD) and Limit of Quantification (LOQ)

Using the peak area for noise (1.51212) and Equation (2), the detection and quantification limits were calculated:LOD = (3 × 1.51212 + 56.361)/2.4266 = 25.09 μg/mL(8)
LOQ = (10 × 1.51212 + 56.361)/2.4266 = 29.46 μg/mL(9)

#### 3.2.3. Precision and Accuracy

To examine the system precision, the same solution containing M-HCl at the concentration of interest (1000 μg/mL) was injected six times to obtain as many chromatograms. From the chromatograms obtained, the peak areas were measured, calculating the mean value, the SD, and the RSD%. The obtained RSD% value is less than 2% (Table 6).

The repeatability and the intermediate precision were evaluated by injecting three different concentrations (700, 1000, and 1300 μg/mL) of M-HCl. For repeatability (intra-day) determination, sets of three replicates of the three concentrations were analyzed on the same day. For intermediate precision (inter-day), three replicates were analyzed on three different days. In both cases, the RSD% value is less than 2%, indicating that the method was precise.

In order to estimate the accuracy, this was determined by performing the recovery experiments for three different solutions at three concentration levels (700, 1000, and 1300 μg/mL), and the final recovery was calculated. The mean recovery is 99.75% in the range 98.12–100.93%.

The precision and the accuracy of the method are presented in Table 7.

Because, in addition to the quantitative determination of M-HCl from tablets, the application of the method to dissolution studies is also considered, the accuracy was also studied at lower concentrations, being calculated in terms of percent relative error (RE%) using Equation (3). Keeping in account the fact that the calculated value for LOQ is 29.46 μg/mL, the accuracy was thus calculated for the range 40–2000 μg/mL, with the mean values of three replicates (Table 8). Since the mean recovery is 99.12% in the range 96.37–101.04%, results show that the method is accurate over the entire measured range. Analyzing the data presented in Table 8, it is observed that the mean percent error (%RE) calculated with Equation (3) is 0.88%, and recovery values lower than 97% are obtained in the case of concentrations lower than 80 μg/mL, which represents less than 8% of the concentration of interest (1000 μg/mL) and, as a result, presents an acceptable variation in the case of dissolution studies.

### 3.3. Stability Studies for M-HCl Solutions

Solutions with concentration of 1000 and 2000 μg/mL, respectively, were analyzed under the method conditions at intervals of 0, 24, and 48 h after preparation, with the solutions being stored in refrigerator, at room temperature, and in a water bath at 37 °C. By integrating the obtained chromatograms, the area of the peaks (Table 9) corresponding to M-HCl were determined, then, using the equation of the regression line, the concentration in M-HCl was calculated.

From this study, it is found that as the temperature at which the aqueous M-HCl solution is stored is increased, its stability is maintained. Thus, for the concentration of 1000 μg/mL, the recovery decreases by 0.19, 0.3, and 0.35% in 24 h and by 0.58, 1.03, and 1.61%; in the case of the concentration of 2000 μg/mL, the recovery decreases by 0.17, 0.28, and 0.37% in 24 h and by 0.88, 1.19, and 1.59% if the solution is stored in the refrigerator, at room temperature, or in a water bath, respectively. The best stability is the solution stored in the refrigerator, and as the storage temperature increases, the stability decreases. In conclusion, the method can be applied to determine the M-HCl content of tablets, with the samples being able to be analyzed within a reasonable time. In the case of dissolution studies, due to the large number of samples, the samples collected at different time intervals are stored in the refrigerator and are analyzed chromatographically in a maximum of 24 h.

### 3.4. Determination of M-HCl Content in Tablets

Under the method conditions, the peak of M-HCl has a retention time around of 5.27 min. Confirmation of the identification of the corresponding M-HCl peak was performed by comparing the retention time of the M-HCl peak in the sample chromatogram with that in the chromatogram of a standard. To increase the quality of the identification, the absorption spectrum measured at the apex of the peak from the sample chromatogram was compared with the standard spectrum of M-HCl stored in the spectra library.

In order to determine the presence of compounds with similar retention times (co-eluents as impurities or excipients), the study of the spectral purity of the chromatographic peak was performed. From this study, it is found that no co-eluents appear, the purity of the peak being higher than 99%.

The experimental results obtained by this method of recovery of M-HCl from different types of tablets are presented in Table 10. The percent content in tablets is calculated using Equation (4).

Analyzing the data in Table 10, it is found that all the pharmaceutical products analyzed have a percentage content of M-HCl in the range 97.78–99.20%, for a permissible deviation of ±5%.

### 3.5. Dissolution Studies

Conventional-release tablets (850 mg tablets and 1000 mg film-coated tablets) were used in the dissolution test for tablets, but also for modified-release tablets (500 mg and 1000 mg prolonged-release tablets).

In the case of conventional release tablets, at the dissolution test performed at simulated gastric fluid with pH = 1.2, there is a minimum release for M-HCl for CP-2 (63.85%), CP-3 (33.8%), and for CP-4 (47.97%) at the first sampling at 5 min. After 30 min, all three formulas of conventional release tablets released over 80% of the declared amount of active substance (Figure 6), which confirms the inclusion of the formulations analyzed in the conventional-release solid dosage forms category according to “2.9.3. Dissolution test for solid dosage forms” and “5.17. Recommendations on methods for dosage forms testing” from European Pharmacopoeia, 10th edition [51]. At the end of the dissolution test, after 2 h, a release of over 98% was registered for all the studied formulations (Figure 6).

In the case of prolonged-release tablets with 500 mg and 1000 mg M-HCl, it is observed that at the first sampling, simulated gastric fluid with pH = 1.2 has a release of M-HCl from the CP-1 sample of 41.74% and in the case of the CP-5 sample of 23.57%, which corresponds to the acceptance criteria of the European Pharmacopoeia, 10th edition [51], by preventing the occurrence of “dose dumping”. After 2 h, in the simulated gastric fluid with pH = 1.2, formulation CP-1 has a maximum release of 49.71%, respectively, 35.96% in CP-5 (Figure 7).

The second specification point of the dissolution test will be defined by the release of approximately 50% of M-HCl from prolonged-release tablets. In the case of the first studied formulation, a much faster release profile is observed, so that the second point is achieved after 2 h in simulated gastric fluid with pH = 1.2 (CP-1: 49.71%), unlike formulation CP-5, which released 53.17% of M-HCl at 4 h in simulated intestinal fluid, pH = 6.8. The final specification point stipulated in the conditions of European Pharmacopoeia is the assurance of complete release, which is understood to be at least 80% achieved in a simulated intestinal fluid pH = 6.8. For the CP-1 test, this point is reached after 5 h (81%), with a maximum release of 100.04% at the end of the test; for the CP-5 81.21% is released after 8 h, with a maximum 99.87% after 24 h (Figure 7). The two samples analyzed are compliant and fall into the release profile for extended-release dosage forms, according to “2.9.3. Dissolution test for solid dosage forms” and “5.17. Recommendations on methods for dosage forms testing” from European Pharmacopoeia, 10th edition [51].

## 4. Conclusions

The paper describes the development of an HPLC method for the determination of M-HCl by the QbD approach using a central composite design by studying the interrelationships of two factors regarding the mobile phase (the pH of aqueous phase and the ratio between acetate buffer and methanol) at three different levels. The optimized CMPs were found after calculations of second-order polynomial exploring quadratic response surfaces for retention time, peak area, and symmetry factor as functions of buffer pH and ratio between buffer and methanol. The optimized solution showed the mobile phase composition containing a mixture of 0.02 M acetate buffer (pH = 3)/methanol in a ratio of 70/30 (*v*/*v*), a case for which the desirability function was very close to 1 (desirability function = 0.9807).

The optimized method was fully validated according to ICH Q2 (R1) Guidelines. The method showed good linearity (r^2^ = 0.9999) in the studied range (10–2000 μg/mL) with good detection and quantification limits (LOD = 25.09 μg/mL and LOQ = 29.46 μg/mL, respectively), good precision (RSD < 2%) and accuracy (mean recovery = 99.75%) in the range 700–1300 μg/mL, and a mean percent error of 1.07% in the range of 40–2000 μg/mL. From the obtained results, it can be concluded that the M-HCl solutions are stable enough at room temperature 24 h after preparations. If the samples were analyzed after more than 24 h, it is recommended that they be kept in a refrigerator.

The optimized and validated method was used for the M-HCL assay from tablets and for dissolution studies. In the case of assay studies, all the pharmaceutical products analyzed have a percentage content of M-HCl in the range 97.78–99.20%, for a permissible deviation of ±5%. In the case of dissolution studies, for conventional-release tablets, after 2 h a release of over 98% was registered for all the studied formulations. In the case of prolonged-release tablets, the two samples analyzed are compliant and fall into the release profile for extended-release dosage forms, according to “2.9.3. Dissolution test for solid dosage forms” and “5.17. Recommendations on methods for dosage forms testing” from European Pharmacopoeia, 10th edition.

## Figures and Tables

**Figure 1 pharmaceutics-14-01187-f001:**
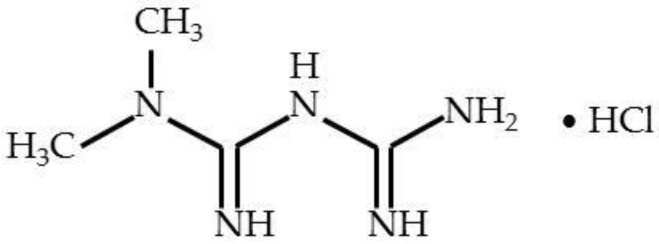
Chemical structure of metformin hydrochloride (M-HCl).

**Figure 2 pharmaceutics-14-01187-f002:**
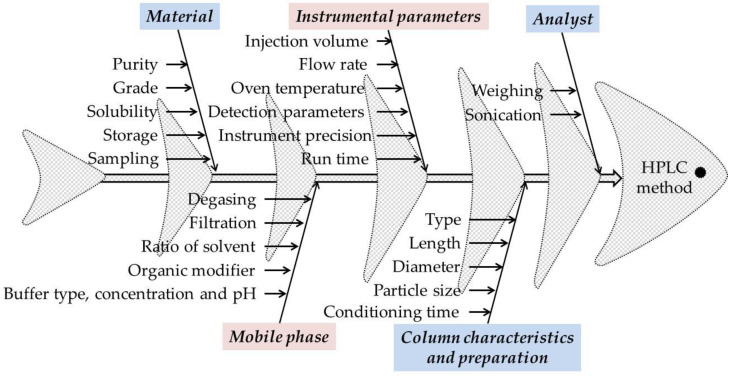
The Ishikawa (fishbone) diagram to identify potential variables in HPLC method development.

**Figure 3 pharmaceutics-14-01187-f003:**
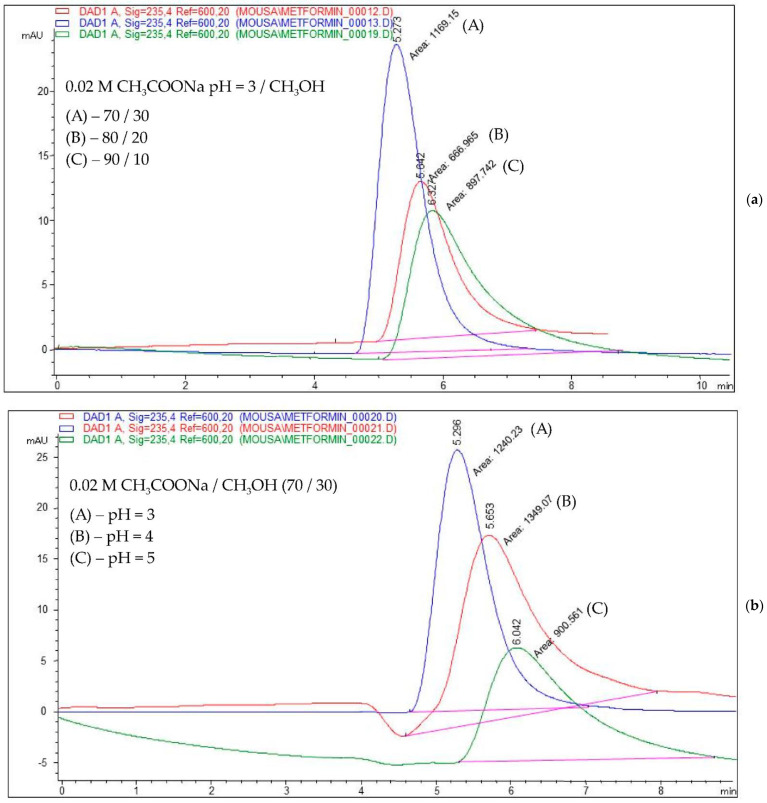
Overlayed chromatograms for a mobile phase consisting of a mixture of (**a**) 0.02 M acetate buffer (pH = 3)/methanol in a ratio of 70/30, 80/20, and 90/10, respectively, *v*/*v*; (**b**) 0.02 M acetate buffer (pH = 3, 4, and 5, respectively)/methanol in a ratio of 70/30, *v*/*v*.

**Figure 4 pharmaceutics-14-01187-f004:**
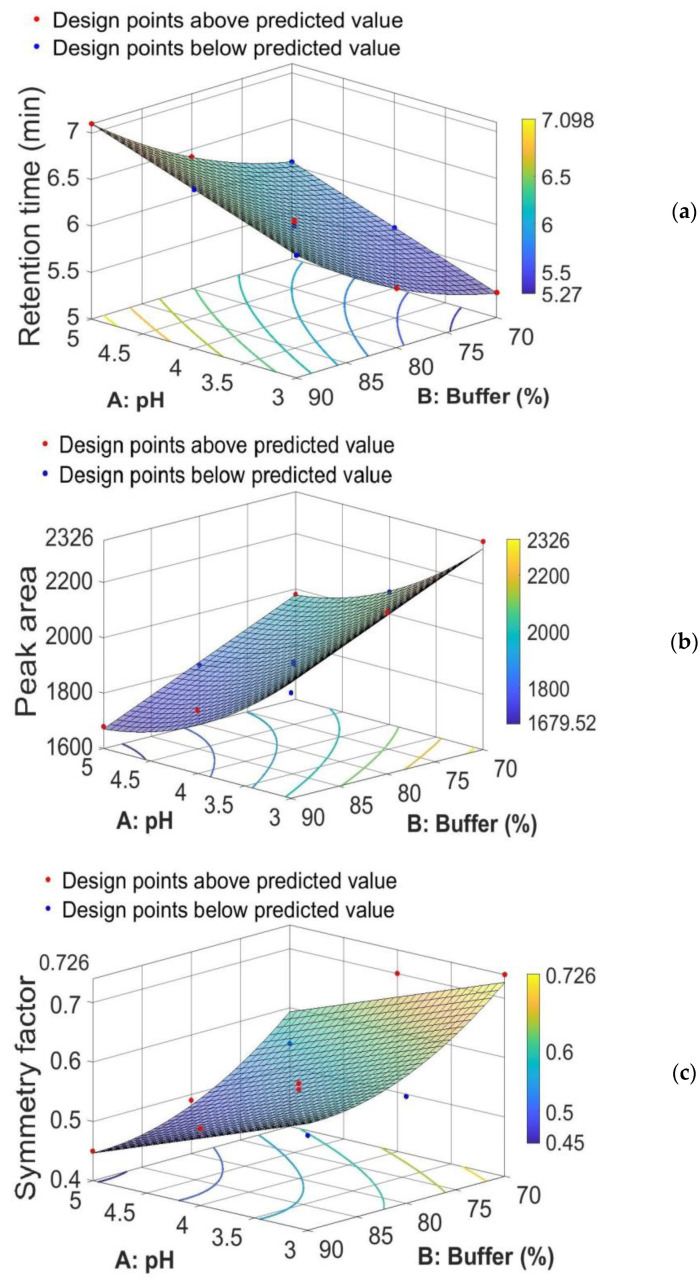
Three-dimensional (3D) response surface plot for (**a**) retention time, (**b**) peak area, and (**c**) symmetry factor, showing effect of % of buffer and pH in the mobile phase.

**Figure 5 pharmaceutics-14-01187-f005:**
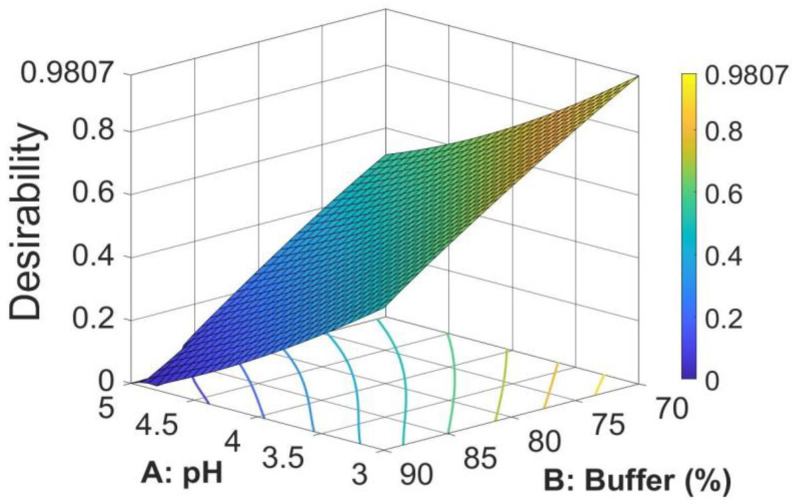
3D surface plot for desirability for optimal formulation.

**Figure 6 pharmaceutics-14-01187-f006:**
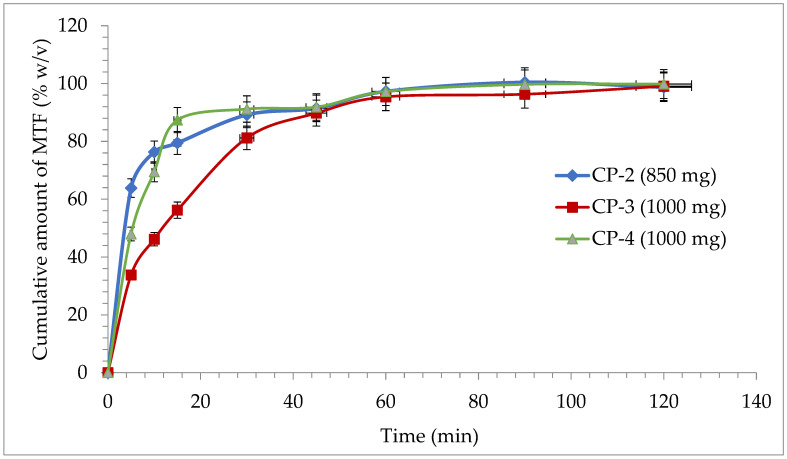
Release profile of M-HCl from conventional tablets.

**Figure 7 pharmaceutics-14-01187-f007:**
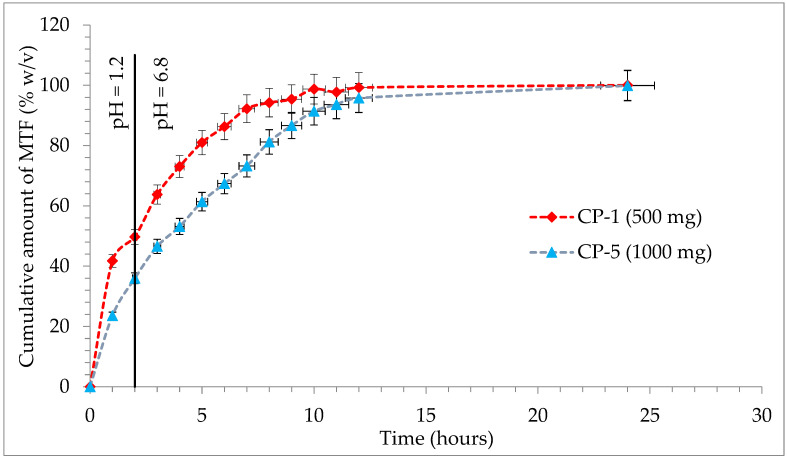
Release profile of M-HCl from prolonged-release tablets.

**Table 1 pharmaceutics-14-01187-t001:** Design matrix as per central composite design (CCD) for optimization of the HPLC method of metformin hydrochloride (M-HCl).

Run	Run Order	Coded Factor Level	
		Factor A	Factor B
1	7	−1	−1
2	9	−1	0
3	4	−1	1
4	8	0	−1
5	5	0	0
6	1	0	0
7	2	0	0
8	6	0	1
9	3	1	−1
10	11	1	0
11	10	1	1
	Level of Factor		
Parameter	Low (−1)	Intermediate (0)	High (+1)
A: Buffer pH	3	4	5
B: Buffer content (%)	70	80	90

**Table 2 pharmaceutics-14-01187-t002:** Design matrix as per the CCD for optimization of parameters for analysis of M-HCl.

Run	Run Order	Factor		Response-1	Response-2	Response-3
		pH	Buffer %	Retention Time	Peak Area	Symmetry Factor
1	7	3	70	5.27	2352.14	0.74
2	9	3	80	5.64	2183.85	0.58
3	4	3	90	6.32	1981.89	0.56
4	8	4	70	5.65	2079.55	0.70
5	5	4	80	6.03	1907.56	0.55
6	1	4	80	5.99	1911.25	0.56
7	2	4	80	6.05	1915.15	0.56
8	6	4	90	6.71	1827.62	0.53
9	3	5	70	6.04	1980.39	0.54
10	11	5	80	6.42	1813.09	0.49
11	10	5	90	7.10	1679.52	0.45

**Table 3 pharmaceutics-14-01187-t003:** Statistical calculation of a second-order polynomial exploring quadratic response surfaces with equation Y = β_0_ + β_1_A + β_2_B + β_3_AB + β_4_A^2^ + β_5_B^2^.

	Retention Time			Peak Area			Symmetry Factor		
	Coefficient	t-Stat	*p*-Value	Coefficient	t-Stat	*p*-Value	Coefficient	t-Stat	*p*-Value
β_0_	10.209	15.766	1.0004 × 10^−6^	4979.6	16.805	6.4667 × 10^−7^	3.876	3.2031	0.015
β_1_	0.38833	56.702	1.3921 ×10^−10^	−736.18	−5.1359	0.0013449	−0.0667	−5.2088	0.0012409
β_2_	−0.19623	−12.063	6.1394 × 10^−6^	−15.384	−12.78	4.1615 × 10^−6^	−0.0692	−3.2763	0.03695
β_4_	---	---	---	70.254	3.9349	0.0056404	---	---	---
β_5_	0.0015567	15.324	1.2146 × 10^−6^	---	---	---	0.00038667	3.0368	0.041091

All coefficients (β_0_–β_5_) are statistically significant at 95% confidence level.

**Table 4 pharmaceutics-14-01187-t004:** Obtained solution for optimized formulation.

pH	Buffer/Methanol	Retention Time (min)	Peak Area	Symmetry Factor	Desirability
3	70/30	5.27	2326.47	0.73	0.9807

**Table 5 pharmaceutics-14-01187-t005:** Final chromatographic conditions.

Parameters	Values
Stationary phase (column)	Thermoscientific ODS Hypersyl^TM^ chromatographic column; (250 × 4.6 mm, 5 μm)
Mobile phase	0.02 M acetate buffer (pH = 3)/methanol (70/30, *v*/*v*)
Flow rate (mL/min)	1
Column temperature	35 °C
Injection volume (μL)	20
Detection wavelength (nm)	235

**Table 6 pharmaceutics-14-01187-t006:** System precision.

No.	Peak Area	Statistics
1	2369.7	Mean = 2376.7 SD = 7.5048 RSD = 0.3158%
2	2380.0	
3	2376.1	
4	2369.6	
5	2389.7	
6	2375.3	

**Table 7 pharmaceutics-14-01187-t007:** Precision and accuracy of the assay method of M-HCl by HPLC.

M-HCl (μg/mL)	Method Precision		Intermediate Precision		Method Accuracy	
	Peak Area	Statistics	Peak Area	Statistics	Peak Area	Recovery (%)
700	1612.3	Mean = 1617.00	1623.9	Mean = 1615.30	1625.8	99.03
	1622.0	SD = 4.8570	1612.2	SD = 7.5439	1610.4	98.12
	1616.7	RSD = 0.3004%	1609.8	RSD = 0.4670%	1628.1	99.17
1000	2355.4	Mean = 2357.97	2362.8	Mean = 2366.63	2385.1	100.61
	2368.8	SD = 9.8053	2381.1	SD = 12.9817	2379.3	100.37
	2349.7	RSD = 0.4158%	2356.0	RSD = 0.5485%	2334.8	98.54
1300	3113.5	Mean = 3112.00	3106.5	Mean = 3137.03	3122.7	100.78
	3104.4	SD = 6.9721	3134.4	SD = 31.9315	3127.6	100.93
	3118.1	RSD = 0.2240%	3170.2	RSD = 1.0179%	3104.8	100.21
Statistical data					Mean recovery = 99.75 Minimum (%) = 98.12 Maximum (%) = 100.93	

**Table 8 pharmaceutics-14-01187-t008:** Accuracy of the method in the range 40–2000 μg/mL.

No.	C_M_ (μg/mL)	Mean Peak Area (n = 3)	C_R_ (μg/mL)	Recovery (%)	RE (%)
1	40	37.50	38.68	96.70	3.30
2	60	83.94	57.82	96.37	3.63
3	80	135.60	79.11	98.89	1.11
4	100	183.50	98.85	98.85	1.15
5	200	416.40	194.82	97.41	2.59
6	300	673.73	300.87	100.29	0.29
7	400	889.07	389.61	97.4	2.60
8	500	1157.43	500.20	100.04	0.04
9	600	1396.60	598.76	99.79	0.21
10	700	1628.10	694.17	99.17	0.83
11	800	1879.00	797.56	99.7	0.31
12	900	2125.00	898.94	99.88	0.12
13	1000	2352.00	992.48	99.25	0.75
14	1100	2610.20	1098.89	99.9	0.10
15	1200	2885.80	1212.46	101.04	1.04
16	1300	3108.97	1304.43	100.34	0.34
17	2000	4797.90	2000.44	100.02	0.02
Recovery (minimum = 96.37%, mean = 99.12%, maximum = 101.04%), mean percent error = 1.07%					

C_M_—real concentration of M-HCl; C_R_—measured concentration of M-HCl.

**Table 9 pharmaceutics-14-01187-t009:** Experimental results obtained in the study of the stability over time of M-HCl solutions with concentrations of 1000 and 2000 µg/mL.

Time (Hours)	Refrigerator (2–8 °C)			Room Temperature (20–25 °C)			37 °C		
	Peak Area	CC (µg/mL)	R (%)	Peak Area	CC (µg/mL)	R (%)	Peak Area	CC (µg/mL)	R (%)
**1000 µg/mL**									
0	2324.4	981.11	98.11	2324.4	981.11	98.11	2324.4	981.11	98.11
24	2319.8	979.21	97.92	2317.2	978.14	97.81	2315.8	977.57	97.76
48	2310.4	975.34	97.53	2299.4	970.81	97.08	2285.4	965.04	96.50
**2000 µg/mL**									
0	4748.6	1980.12	99.01	4748.6	1980.12	99.01	4748.6	1980.12	99.01
24	4740.4	1976.74	98.84	4735.3	1974.64	98.73	4730.6	1972.70	98.64
48	4706.1	1962.61	98.13	4690.8	1956.30	97.82	4671.8	1948.47	97.42

CC—calculated concentration, R (%)—recovery in percent.

**Table 10 pharmaceutics-14-01187-t010:** Experimental data obtained from the study of the M-HCl content of tablets.

Pharmaceutical Product	Pharmaceutical Form	Country Manufacturer	*A* (mg)	*M_m_* (g)	*a* (g)	*P_A_*	C (%)
CP–1	prolonged-release tablet	France	500	0.727	0.7281	2339.56	98.59
CP–2	tablet	Romania	850	1.0146	0.5962	2348.43	99.20
CP–3	film-coated tablet	France	1000	1.1238	0.5705	2352.63	97.78
CP–4	film-coated tablet	Germany	1000	1.1065	0.553	2332.33	98.48
CP–5	prolonged-release tablet	France	1000	1.4549	0.7285	2348.18	98.95

*A*—declared content (mg), *M_m_*—average weight of a tablet (g), *a*—tablet powder (g), *P_A_*—peak area, C%—percent content in tablet, *S* = 2.4266: slope of the calibration curve, *Int* = −56.361: intercept of the calibration curve.

## Data Availability

Not applicable.

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
