# Peer review of "Implementation of QbD Approach to the Analytical Method Development and Validation for the Estimation of Metformin Hydrochloride in Tablet Dosage Forms by HPLC"

_pharmaceutics, 2022, doi:10.3390/pharmaceutics14061187_

Round 1

Reviewer 1 Report

Interesting article

Author Response

To see the authors' response, go to the attached file.

Reviewer 2 Report

Dear authors, your article "Implementation of QbD Approach to the Analytical Method 2 Development and Validation for the Estimation of Metformin 3 Hydrochloride in Tablet Dosage Forms by HPLC" overall is well written, well discussed, and step by step. But before its final acceptance, I have some issues which needed to ab addressed to make the article more attractive for its final shape.

  1. Kindly arrange the pixels of the HPLC chromatogram of Figure 8. It is very blurry. Nothing is visible inside the chromatogram.
  2. The intro section is very very long kindly adjust it.
  3. The results and Discussion are two sections but the authors merged them so in some sections some work is discussed but in some areas not. So I strongly suggest authors have a separate section of discussion after the results section.
  4. For the research article 82 references are really a lot. KIndly try to minimize them.

Author Response

(The authors gave the same response as above.)

Reviewer 3 Report

I am hesitant to choose a rating for the 'overall merit' of the manuscript. While the presentation of the QbD is well done by accompanying a relatively straightforward application, the manuscript seems to be written by two people (or one person with very different levels of understanding). The chromatography part seems to be written by a student in a report. There are a lot of details (some irrelevant, some relevant but incredibly detailed for a publication). The figures illustrating the surface methodology are small so the reader cannot follow the explanations, while some other figures (7, 8 , 9) are not necessary (not even for the supplementary).

While I consider the work suitable for publication, the manuscript in its current for is doing it disservice. I urge the authors to reconsider the details included, the structure of the information presented, to align the results and discussion with the aim, and definitely the length of the text.

Author Response

(The authors gave the same response as above.)

Reviewer 4 Report

The article is well structured and organized. It shows the importance of quality by design tool for development of HPLC method for Metformin HCl determination. It is interesting article  in which Design of experiments was used for the multivariate optimization of experimental conditions of HPLC method. However, few formatting mistakes were noticed. I recommend it for publication after going through to eliminate minor formatting mistakes.

Author Response

(The authors gave the same response as above.)
